# Globospiramine Exhibits Inhibitory and Fungicidal Effects against *Candida albicans* via Apoptotic Mechanisms

**DOI:** 10.3390/biom14060610

**Published:** 2024-05-22

**Authors:** Joe Anthony H. Manzano, Simone Brogi, Vincenzo Calderone, Allan Patrick G. Macabeo, Nicanor Austriaco

**Affiliations:** 1The Graduate School, University of Santo Tomas, España Blvd., Manila 1015, Philippines; joeanthony.manzano.gs@ust.edu.ph; 2UST Laboratories for Vaccine Science, Molecular Biology and Biotechnology, Research Center for the Natural and Applied Sciences, University of Santo Tomas, España Blvd., Manila 1015, Philippines; naustriaco@ust.edu.ph; 3Laboratory for Organic Reactivity, Discovery, and Synthesis (LORDS), Research Center for the Natural and Applied Sciences, University of Santo Tomas, España Blvd., Manila 1015, Philippines; 4Department of Pharmacy, University of Pisa, Via Bonanno 6, 56126 Pisa, Italy; vincenzo.calderone@unipi.it; 5Department of Chemistry, College of Science, University of Santo Tomas, España Blvd., Manila 1015, Philippines; 6Department of Biological Sciences, College of Science, University of Santo Tomas, España Blvd., Manila 1015, Philippines

**Keywords:** globospiramine, bisindole alkaloid, *Voacanga globosa*, *Candida albicans*, *Candida tropicalis*, molecular docking, molecular dynamics, antifungal, apoptosis

## Abstract

Candidiasis is considered an emerging public health concern because of the occurrence of drug-resistant *Candida* strains and the lack of an available structurally diverse antifungal drug armamentarium. The indole alkaloid globospiramine from the anticandidal Philippine medicinal plant *Voacanga globosa* exhibits a variety of biological activities; however, its antifungal properties remain to be explored. In this study, we report the in vitro anticandidal activities of globospiramine against two clinically relevant *Candida* species (*C. albicans* and *C. tropicalis*) and the exploration of its possible target proteins using in silico methods. Thus, the colony-forming unit (CFU) viability assay revealed time- and concentration-dependent anticandidal effects of the alkaloid along with a decrease in the number of viable CFUs by almost 50% at 60 min after treatment. The results of the MIC and MFC assays indicated inhibitory and fungicidal effects of globospiramine against *C. albicans* (MIC = 8 µg/mL; MFC = 8 µg/mL) and potential fungistatic effects against *C. tropicalis* at lower concentrations (MIC = 4 µg/mL; MFC > 64 µg/mL). The FAM-FLICA poly-caspase assay showed metacaspase activation in *C. albicans* cells at concentrations of 16 and 8 µg/mL, which agreed well with the MIC and MFC values. Molecular docking and molecular dynamics simulation experiments suggested globospiramine to bind strongly with 1,3-β-glucan synthase and Als3 adhesin—enzymes indirectly involved in apoptosis-driven candidal inhibition.

## 1. Introduction

The increased occurrence and severity of fungal infections have greatly contributed to the escalation of disease-associated morbidity and mortality rates, with approximately 1.5 million deaths annually [1,2,3]. Treatment failures are mostly attributed to the emergence and re-emergence of resistant strains, which are among the consequences of the irresponsible use of available antifungal drugs and the innate evolutionary mechanisms of the causative agents against therapeutic pressures [4,5].

Invasive fungal infections are caused by different species, mostly belonging to the genera *Aspergillus*, *Pneumocystis*, *Candida*, and *Cryptococcus*. Among these, *Candida* spp. are the most frequently reported pathogens [6]. *Candida* spp. transition into opportunistic pathogens in immunologically weak and immunocompromised patients, leading to local and systemic infections [7]. However, in the last few decades, there has been an increased incidence of deep fungal infections or chronic candidiasis (long-term infection caused by *Candida* species) including healthy individuals [8,9]. In addition to *C. albicans*, *C. tropicalis*, *C. glabrata*, *C. parapsilosis,* and *C. krusei* infections have been reported in clinical settings. These species have developed mechanisms to ensure efficient colonization and infection of their hosts despite therapeutic pressures. Therefore, this necessitates the discovery of new antifungal agents with novel and/or multitargeting mechanisms [10,11].

The current clinically approved antifungals belong to the following structural classes: polyenes, azoles, and echinocandins [12]. Polyenes like amphotericin B target ergosterol, which is selectively found in fungal cell membranes. Azoles inhibit fungal cytochrome P450-dependent enzymes, resulting in impaired ergosterol synthesis. Echinocandins block the 1,3-β-*D*-glucan synthase, disrupting fungal cell wall biosynthesis [13]. The fact that most of the chemotherapeutic-based treatment armamentarium against candidiasis is limited to these three classes calls for new and diverse drug congeners from different structural classes.

Relevant to our study, indole alkaloids have long been explored for their biological properties. The indole moiety contributes to the efficient binding of compounds to a number of disease targets, thus conferring favorable biological activities. In the context of anticandidal drug discovery, some reported indole alkaloids include kopsifolines A, G, H, I, J, and K from *Kopsia fruticosa* [14], ibogaine [15], and chetomin and chaetoglobinol A from *Chaetomium globosum* [16]. Moreover, 6 out of 27 isolated monoterpene indole alkaloids from the plant *Rhazya strictra* demonstrated inhibitory properties against six *Candida* strains [17]. Other anticandidal indole alkaloids derived from marine organisms include indolepyrazines A and B from *Acinetobacter* sp. ZZ1275 [18]. Thus, indole alkaloids and their derivatives have great potential for development as anticandidal agents. Among the prolific producers of biologically active indole alkaloids of plant origin is the genus *Voacanga* of the family Apocynaceae. In the Philippines, the medicinal plant *Voacanga globosa* (Blanco) Merr. has numerous biological activities [19,20,21], including *C. albicans* inhibition [22]. As part of our research effort to explore antimicrobial agents from Philippine medicinal plants, as well as to identify potential bioactive phytoconstituents that may contribute to the previously reported anticandidal properties of *V. globosa*, we hereby disclose the anticandidal activities of the spirobisindole alkaloid globospiramine (Figure 1) [19] against *C. albicans* and *C. tropicalis* using in vitro and computational approaches (molecular docking and molecular dynamics (MD) simulations). 

## 2. Materials and Methods

### 2.1. Test Compound

Globospiramine with 98% purity was isolated from *Voacanga globosa* as previously reported [19]. The test concentrations of globospiramine were prepared by dilution with dimethyl sulfoxide (DMSO). 

### 2.2. Colony-Forming Unit Viability Assay

A colony-forming unit (CFU) viability assay was performed following previously reported methods [23]. Briefly, *C. albicans* ATCC 90028 and *C. tropicalis* ATCC 750 cells were inoculated in Sabouraud dextrose broth (SDB) and incubated overnight. The resulting culture was required to reach OD_600_ = 0.3 to 0.8 prior to serial dilution until approximately 500 CFUs/mL were attained. The test compound globospiramine in varying concentrations (2.5, 50, and 100 µg/mL) was added to the diluted suspension, and 50 µL was aliquoted from the diluted suspension and spread plated on Sabouraud dextrose agar (SDA) plates after 0, 30, 60, and 90 min. After 48 h of incubation, the percentage CFU viability was measured using the following equation: $$\%\;CFU\;viability=\frac{{CFU}_{x}\;}{{CFU}_{0}}\;\times \;100$$
where ${CFU}_{x}$ is the number of surviving colonies at each specific time point (*x* = 30, 60, or 90 min) for each test concentration of globospiramine, while ${CFU}_{0}$ is the number of colonies at the starting point (time = 0). DMSO (negative/vehicle) and amphotericin B were utilized as controls. The experiment was performed in triplicate. 

### 2.3. MIC and MFC Determination

The minimum inhibitory concentration (MIC) was assessed based on CLSI M27-A3 2008 guidelines on microdilution for yeast species. Fresh yeast suspension (OD_600_ = 0.3 to 0.8) in SDB was diluted to yield 5.0 × 10^2^ to 2.5 × 10^3^ cells/mL. Two-fold serial dilutions from 256 µg/mL to 1 µg/mL were prepared in a 96-well microplate for globospiramine. For the positive control amphotericin B, the concentrations used were 0.125 to 16 µg/mL. Each well contained 100 µL test concentrations, 5 µL standardized yeast suspension in SDB, and 100 µL RPMI 1640 broth medium. The absorbance was recorded using a Glomax Discover Microplate Reader (Promega) after 24 h of incubation. The MIC was recorded as the lowest concentration *to induce a significant decrease in OD_600_ compared to the growth controls*. Tests were performed in triplicate. For the minimum fungicidal concentration (MFC) determination assay, 10 µL samples from the wells containing the MIC and two concentrations higher were spread plated on SDA. The MFC was recorded as the lowest concentration where no colony growth was observed in the three independent plates after 48 h of incubation. 

### 2.4. FAM-FLICA Poly-Caspase Assay

The FAM-FLICA poly-caspase assay was performed according to the manufacturer’s instructions provided in the kit (ImmunoChemistry Technologies, Bloomington, MN, USA) with modifications. The FAM-FLICA reagent has been used in studies to elucidate yeast metacaspase activities [24,25]. For the treatment concentrations, the MIC, 2 × MIC, and 4 × MIC of globospiramine against *C. albicans* and *C. tropicalis* were used. In the 96-well microplate, *Candida* cells were exposed to the treatment groups for 24 h. Each well contained approximately 1 × 10^6^ candidal cells/mL. The FAM-FLICA reagent was then added. After 50 min of incubation at 30 °C, fluorescence readouts were obtained using the Glomax Microplate Discover Reader (Promega GmbH, Walldorf, Germany) (excitation: 490 nm; emission: 530 nm). Three independent experiments were conducted. For statistical analysis, one-way ANOVA was performed, followed by pairwise analysis with DMSO (negative/vehicle control) as the reference (* *p* < 0.05, ** *p* < 0.01, *** *p* < 0.001, and **** *p* < 0.0001).

### 2.5. Molecular Docking against C. albicans Proteins

#### 2.5.1. Ligand and Protein Preparation

Globospiramine was considered as the ligand, with the *C. albicans* target proteins as receptors and/or molecular targets. The following PDB IDs were used: 1EQP (1,3-β-glucan synthase), 4QUV (δ-14-sterol reductase), 5TZ1 (lanosterol 14-alpha demethylase or CYP51), 5UIV (thymidylate kinase), 4LEB (Als3 adhesin), and 2Y7L (Als9-2). These PDB IDs have already served as key targets in other molecular docking studies and/or are considered pharmaceutical targets of current anticandidal drugs [26,27,28]. To prepare the ligand, its structure was drawn in ChemDraw (18.1), optimized in Avogadro (1.2.0), and saved as a .mol2 file. For the protein preparation, non-standard residues were removed in UCSF Chimera (1.17.3), followed by minimization using the steepest descent and conjugate gradient methods. The output file was saved in .pdb format [29].

#### 2.5.2. Molecular Docking and Visualization of Interactions

The prepared ligands and protein targets were combined using UCSF Chimera (1.17.3). Actual docking simulation experiments were carried out using the flexible ligand in a flexible active site protocol based on the BFGS algorithm coupled with AutoDock Vina. Grids were generated to encompass the target binding domains. To visualize the interactions, the output files were processed using BIOVIA Discovery (4.1) [30].

### 2.6. Molecular Dynamics Simulations 

MD simulation experiments were conducted using the Desmond package (Desmond Molecular Dynamics System 6.4 academic version, D. E. Shaw Research (“DESRES”), New York, NY, USA, 2020. Maestro-Desmond Interoperability Tools, Schrödinger, New York, NY, USA, 2020). The three-dimensional ligand/protein complexes (globospiramine within 1,3-β-glucan synthase (PDB ID: 1EQP) and Als3 adhesin (PDB ID: 4LEB)) obtained by molecular docking studies were prepared using the system builder tool, available in Desmond software, to produce suitable complexes for MD simulation studies. Accordingly, the ligand/protein complexes were placed into an orthorhombic box and solvated by water molecules (TIP3P water model) [31,32]. A physiological concentration of monovalent ions (0.15 M) was used by adding Na^+^ and Cl^−^ ions to the biological systems. MD simulation studies were conducted using the OPLS3 force field [33], and calculations were performed utilizing the CUDA API technology on two NVIDIA graphics processing units (GPUs). A constant number of particles, constant temperature (300 K by the Nosé–Hoover thermostat method [34]), and pressure (1.01325 bar by the Martyna–Tobias–Klein method [35]) were considered using the NPT ensemble class. To assess the motion for bonded and non-bonded interactions within the short-range cutoff, the RESPA integrator was adopted (inner time step of 2.0 fs) [36]. To calculate long-range electrostatic interactions (short-range electrostatic interactions were fixed at 9.0 Å), the particle mesh Ewald method (PME) was employed [37]. To equilibrate the biological systems, the default protocol available in Desmond was used. The protocol consists of several constrained minimizations and MD simulations that were applied to each biological system to progressively relax and bring them to equilibrium. The Desmond application’s simulation event analysis tools were utilized to examine the MD results produced throughout the MD simulation calculations, as previously reported [38]. 

## 3. Results

### 3.1. Effects of Globospiramine on C. albicans and C. tropicalis CFU Viability

The initial screening for inhibitory activities of globospiramine against the pathogenic yeast species *C. albicans* and *C. tropicalis*, in terms of CFUs, was conducted in vitro using a CFU viability assay. Globospiramine exhibited time- and concentration-dependent activities against *C. albicans* and *C. tropicalis* by significantly decreasing the percentage of viable CFUs after 30 min, 60 min, and 90 min of exposure. After 60 min of exposure to globospiramine at 2.5 µg/mL, the CFU count for both *Candida* species was reduced to approximately 50% of the original count (Figure 2). It is, however, important to note that although the decrease in % CFU viability seemed to be several folds lower compared to the 100% baseline at time = 0 min, the % CFU viability was not completely 0% because of the surviving colonies observed.

### 3.2. MIC and MFC of Globospiramine versus C. albicans and C. tropicalis

The significant decrease in CFU viability induced by exposure to globospiramine over a short period of time prompted us to identify its MIC and MFC values, which were identified through a standardized method based on CLSI. The MIC and MFC were determined for both *Candida* species to support the CFU viability data using microdilution and spread plate techniques. The lowest concentration that significantly induced decreased OD_600_ compared to growth control was considered as the MIC, whereas MFC was the lowest concentration that completely inhibited the growth of yeast colonies. In agreement with our results on the CFU viability assay, globospiramine showed moderately strong MIC (8 µg/mL) and MFC (8 µg/mL) values against *C. albicans* (Table 1) compared to the positive standard drug control, amphotericin B. Interestingly, a lower concentration of globospiramine was effective against *C. tropicalis* (MIC = 4 µg/mL); however, such activity might be attributed to growth inhibition, or conversely a much higher concentration (MFC > 64 µg/mL) was necessary to promote fungicidal effects (Table 1). 

### 3.3. Apoptosis-Inducing Activities of Globospiramine vs. C. albicans and C. tropicalis

The metacaspase-activating activity of globospiramine versus *C. albicans* and *C. tropicalis* cells was also investigated to determine its possible mechanism of action. Globospiramine induced a significant increase in relative fluorescence units (RFUs) compared to the vehicle control DMSO, triggering metacaspase activation responses in *C. albicans* cells at the 16 (*p* < 0.01) and 8 µg/mL (*p* < 0.05) test concentrations (Figure 3). These concentrations corroborated to the MIC and MFC values in Table 1. Meanwhile, globospiramine did not induce metacaspase activation in *C. tropicalis* cells. This is to be expected since the MFC value against *C. tropicalis* was noted as >64 µg/mL. 

### 3.4. Molecular Docking against C. albicans Targets

To enrich the characterization of globospiramine, we conducted a series of in silico studies to select potential targets related to the observed anticandidal effect. Accordingly, six protein targets previously reported to play significant roles in the pathogenesis of candidiasis caused by *C. albicans*, and potentially related to apoptosis-driven candidal inhibition, were selected for molecular docking experiments. In particular, we selected 1EQP (1,3-β-glucan synthase), 4QUV (δ-14-sterol reductase), 5TZ1 (lanosterol 14-alpha demethylase or CYP51), 5UIV (thymidylate kinase), 4LEB (Als3 adhesin), and 2Y7L (Als9-2). Globospiramine showed the most significant predicted binding affinities to 1,3-β-glucan synthase and Als3 adhesin (Table 2, Figure 4A,B). Compared with the positive controls caspofungin and amphotericin B, our compound showed a more favorable binding energy against all these targets.

### 3.5. Molecular Dynamics Simulations

To improve the reliability of target identification and validate the docking results, we conducted MD simulation experiments on the most promising targets. In particular, the complexes 1,3-β-glucan synthase/globospiramine and Als3 adhesin/globospiramine were considered for MD simulation studies. Figure 5 shows the MD simulation output for the complex 1,3-β-glucan synthase/globospiramine. Based on 100 ns of MD simulation, we observed a general stability of the selected biological system, highlighted by the low RMSD values of the protein and the ligand and by the low RMSF value, indicating small fluctuations in the biological system. Considering the main interactions found by molecular docking studies, we observed that the H-bond established with residue His253 was maintained, although it became water-mediated. In addition, a strong H-bond network was detected with residue Asp227. Other polar contacts, mainly water-mediated with His254, Glu262, Arg265, and Asp280, were observed. The hydrophobic interactions with Phe229, Tyr255, Phe258, and Trp277 were well maintained during the simulation, with additional hydrophobic contacts with Phe232 and His252. 

Regarding the Als3 adhesin/globospiramine complex, the MD simulation results are illustrated in Figure 6. In addition, in this case, we observed a general stability of the biological system with a small fluctuation of the protein, as indicated by the RMSD and RMSF values. Considering the main contacts governing the binding mode of globospiramine within the selected binding site of the Asl3 adhesin, we observed that the H-bond with Thr168 was well maintained, as were the ionic interactions with the residue Asp169. Additional polar contacts that could contribute to stabilizing the binding mode were detected with residues Ala19, Asn22, and Arg294. Hydrophobic interactions with residues Val161, Tyr166, and Leu167 were still evident at low frequencies. More favorable hydrophobic contacts were established from globospiramine with Tyr226 and Trp295. 

## 4. Discussion

Globospiramine is a spirobisindole alkaloid isolated and identified from the Philippine endemic medicinal plant *Voacanga globosa*. It exhibited biological activities such as cholinesterase inhibitory, antiviral, antimycobacterial, and cytotoxic activities [19,20,39]. *V. globosa* extracts on the other hand demonstrated anticancer and antifungal activities [21,22]. Relevant to our study, indole alkaloids are known to exert antifungal activities [14,40]. For example, decreased viability of fluconazole-resistant *C. albicans* was observed upon treatment with Tabernaemontana divaricate indole alkaloids [41]. Other studies have also indicated potential targets of these indole alkaloids, such as isocitrate lyase and other extracellular enzymes implicated in the lipolytic and proteinase activities of *C. albicans* [15,42]. 

Among the emerging targets in anticandidal drug discovery, metacaspase activation is a promising target because of its significant and highly regulated role in apoptosis. Herein, we report the metacaspase-activating potential of globospiramine against *C. albicans* and *C. tropicalis* cells using the FAM-FLICA poly-caspase assay. Yeast and mammalian apoptosis share markers such as DNA fragmentation, metacaspase activation, and reactive oxygen species (ROS) accumulation. FAM-FLICA binds to the yeast metacaspase and thus used to assess yeast apoptosis [24,25,43,44]. The antifungal activities of natural products have been investigated in the context of apoptosis induction as a mechanism of their fungicidal action [45,46,47]. Our results indicate that the fungicidal effect of globospiramine against *C. albicans* occurs through metacaspase activation leading to apoptosis, which may also explain the correlation of MIC, MFC, and metacaspase-inducing activity at the same test concentration. Meanwhile, against *C. tropicalis*, globospiramine may be fungistatic and not fungicidal, although other possible modes of cell death like necrosis may be investigated. Therefore, further investigation of the effect of globospiramine in promoting the growth inhibition of *C. tropicalis* cells, as well as in other upstream apoptotic pathways for *C. albicans*, is warranted. Thus, our study indicates for the first time that the *V. globosa* phytoconstituent globospiramine has anticandidal activity and could be responsible for the purported antifungal activity of the medicinal plant.

Our work determined the mechanism underlying the growth inhibition of *C. albicans* at the cellular level, characterizing the activation of the apoptotic pathway in *C. albicans* as the main mechanism of globospiramine in significantly reducing the growth of the microorganism. Furthermore, we conducted in silico studies to enlarge the investigation of globospiramine, probing the potential effects of the compound at the molecular level. We analyzed the possible drug targets of the compound, considering the different targets related to the observed pro-apoptotic effects. Among the selected targets, the in silico analysis showed that globospiramine could target 1,3-β-glucan synthase and Als3 adhesin. The results were confirmed by MD simulation, which clearly indicated that globospiramine could bind to the target proteins. The first potential target, 1,3-β-glucan synthase, is an enzyme important for fungal cell wall synthesis. This enzyme facilitates the creation of β(1→3) glycosidic bonds within 1,3-β-glucan molecules, utilizing uridine diphosphate-activated glucose (UDP-Glc) as the source of sugar and transporting the resulting glucan across the membrane [48]. In general, echinocandins like caspofungin are known to elicit inhibitory effects on this enzyme in the plasma membrane. Unstable and impaired fungal cell walls result in morphogenic and intracellular changes that cause cell death [49,50]. In addition, the fact that fungal cell walls and the enzyme itself are absent in human cells makes it more ideal as a therapeutic target. Recently, this enzyme has served as a key target among newly discovered compounds and other antifungals [51,52]. Drug-induced damage to the *C. albicans* cell membrane and cell wall upon inhibition of this enzyme was also reported to result in cellular stress, subsequent apoptosis, and G0/G1 cell cycle arrest [53]. However, there are a myriad of reports on phenotypic changes in the enzyme caused by gene mutations in the coding region of *FKS1*, a gene involved in the biosynthesis of 1,3-β-glucan synthase. As a result, such mutations promote echinocandin resistance [54,55]. Therefore, the discovery of multitargeting agents against 1,3-β-glucan synthase and other virulence factors and/or molecular entities is deemed a logical strategy. 

The second potential target investigated is Als3 adhesin. The *ALS* gene family encodes cell surface proteins in *C. albicans*. These proteins function in adhesion to host cells and various surfaces and are thus implicated in biofilm formation. Biofilms are important determinants in yeast infections and are directly associated with drug failure and resistance. Interestingly, Als proteins, including Als3 adhesin, are not expressed in human cells [56,57,58]. Therefore, Als3 is considered an emerging, pathogenetically relevant target for new-generation antifungals against *Candida* species [59]. Computational studies suggested globospiramine as potential inhibitor of these enzymes with comparable predicted affinity. Accordingly, a potential multitargeting antagonistic effect of globospiramine on *C. albicans* could be hypothesized. The inhibition of one or both of the identified enzymes can be directly or indirectly related to necrosis and/or apoptosis, respectively, as in the case of the anticandidal drug caspofungin, which exhibits concentration-dependent mechanisms against *C. albicans*, inhibiting 1,3-β-glucan synthesis, promoting apoptosis, and inducing necrosis [60]. Previous studies have indicated the role of these enzymes in apoptosis. For example, 1,3-β-glucan synthase is involved in cell wall synthesis; therefore, an inhibitor may potentially interrupt the normal biosynthetic pathway to polymerize the candidal cell wall, leading to ROS hyperaccumulation—a hallmark of apoptosis [61]. The rapid killing of *C. albicans* by certain sets of antimicrobial peptides may be due to apoptosis preceded by cell wall disruption and ROS accumulation [62]. A lipopeptide has also been reported to inhibit *C. albicans* growth by directly impairing the fungal cell wall, leading to an increase in ROS levels and mitochondrial dysfunction, which can activate apoptotic pathways [63]. The inhibition of Als3 triggers stress signals and responses due to the loss of adhesion-dependent survival signals in *C. albicans*. These stress signals can activate the apoptotic machinery via oxidative stress, DNA damage, or mitochondrial impairment. Additionally, biofilm formation may be disrupted upon inhibition of this protein. In fact, apoptosis has been reported in *Candida* biofilms under the therapeutic pressure of amphotericin B [46,64]. 

Although favorable findings were reported in this study, certain limitations need to be considered. In vitro validation of our in silico results is recommended to further confirm the molecular targets of globospiramine in candidal infection pathophysiology. The results of both molecular docking and MD studies provided reliable predictions of how globospiramine could bind to potential target proteins involved in the activation of the apoptotic pathway. Further experiments are recommended to confirm the pro-apoptotic, fungicidal, and enzyme-inhibitory concentrations. Nevertheless, our computational data narrowed down the potential therapeutic targets of globospiramine in the pathophysiology of *C. albicans* infections.

## 5. Conclusions

Overall, our study reports the fungicidal potential of the spirobisindole alkaloid globospiramine from the Philippine medicinal plant *Voacanga globosa*, particularly against *Candida albicans* via apoptotic-aided mechanisms. An in silico investigation indicated two potential enzymes (1,3-β-glucan synthase and Als3 adhesin) that could be targeted by globospiramine to exert the observed inhibitory effect on *C. albicans* growth. Our in vitro findings also demonstrated the fungistatic effects of globospiramine against *C. tropicalis*. Accordingly, globospiramine could represent a good biomolecular candidate for exploring and discovering anticandidal drugs with improved therapeutic effects. 

## Figures and Tables

**Figure 1 biomolecules-14-00610-f001:**
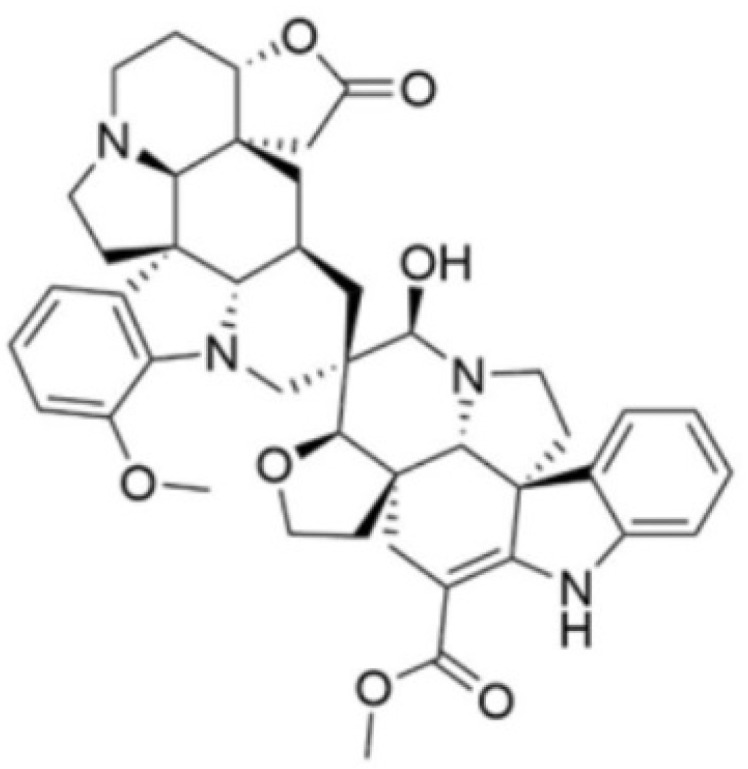
Structure of globospiramine.

**Figure 2 biomolecules-14-00610-f002:**
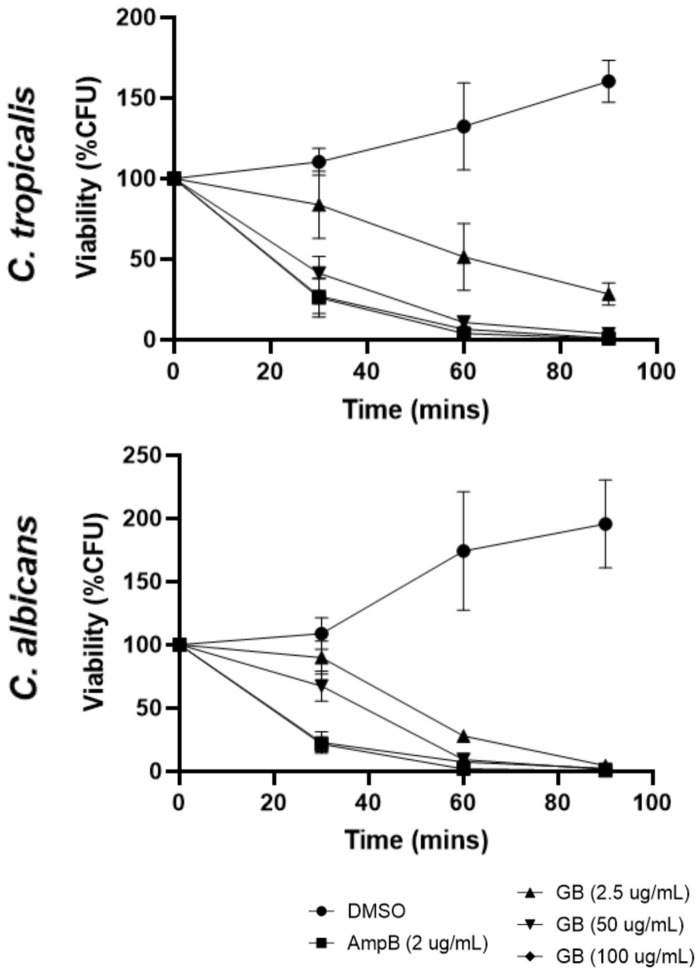
Time- and concentration-dependent effects of globospiramine on the CFU viability of *C. albicans* and *C. tropicalis*.

**Figure 3 biomolecules-14-00610-f003:**
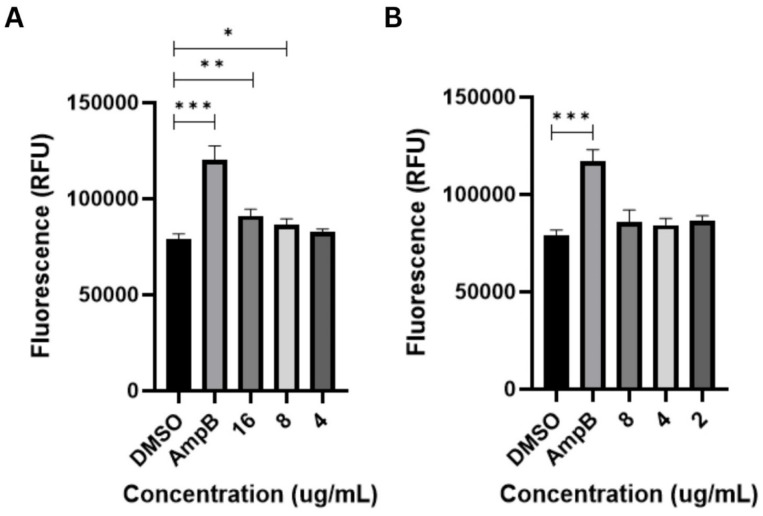
The spirobisindole alkaloid globospiramine significantly induced apoptosis in (**A**) *C. albicans* cells at 16 and 8 µg/mL concentrations, which agreed with its MIC and MFC. However, these effects were not observed in (**B**) *C. tropicalis* cells (* *p* < 0.05, ** *p* < 0.01, and *** *p* < 0.001). DMSO was used as negative control and amphotericin B (AmpB) (0.5 µg/mL) as positive control.

**Figure 4 biomolecules-14-00610-f004:**
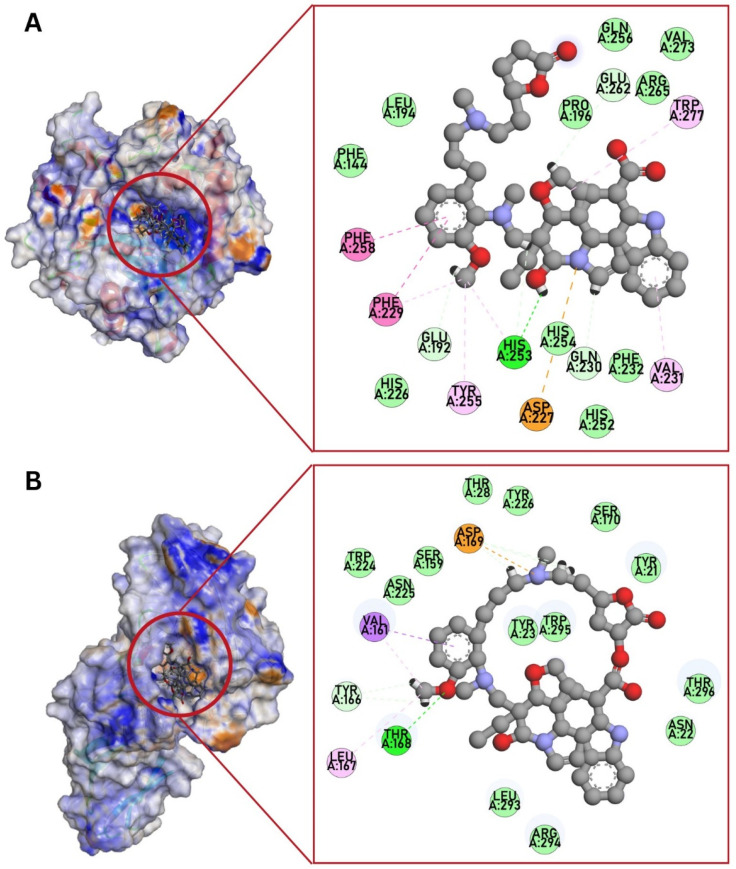
Potential binding mode of globospiramine within the binding site of (**A**) 1,3-β-glucan synthase and (**B**) Als3 adhesin.

**Figure 5 biomolecules-14-00610-f005:**
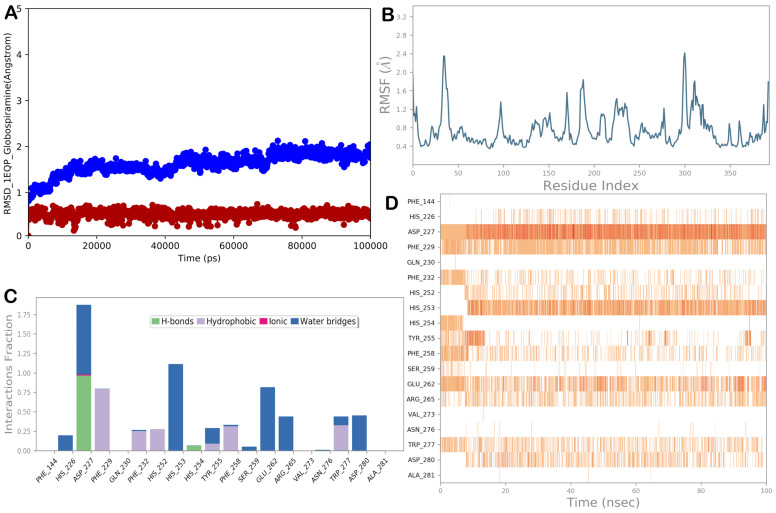
(**A**) RMSD evaluation (protein: blue line; ligand: red line). (**B**) RMSF assessment for the complex 1,3-β-glucan synthase (1EQP)/globospiramine, obtained by docking studies, following a 100 ns MD simulation. (**C**,**D**) Globospiramine observed throughout the MD run. Four types of interactions can be distinguished: water bridges (blue), ionic (magenta), hydrophobic (grey), and H-bonds (green). Over the trajectory, the stacked bar charts are normalized. For instance, a value of 0.7 indicates that a particular contact was maintained 70% of the time during simulation. Values greater than 1.0 could occur because a protein residue could interact with the ligand more than once using the same subtype. A timeline explanation of the primary interactions is shown in the following diagram in the figure. Those residues that interacted with the ligand in each trajectory frame are displayed in the output. A darker orange hue denotes several contacts that some residues had with the ligand. The Maestro and Desmond software tools were utilized to generate the pictures (Maestro, Schrödinger LLC, release 2020-3).

**Figure 6 biomolecules-14-00610-f006:**
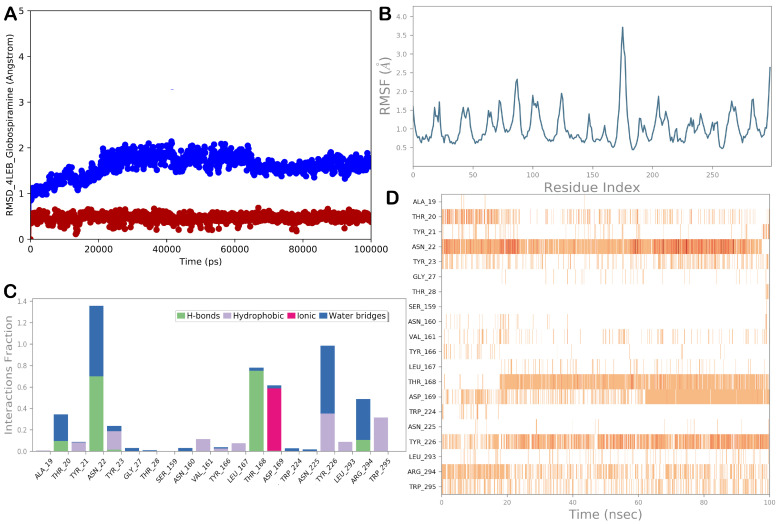
(**A**) RMSD evaluation (protein: blue line; ligand: red line). (**B**) RMSF assessment for the complex Als3 adhesin (4LEB)/globospiramine, obtained by docking studies, following a 100 ns MD simulation. (**C**,**D**) Globospiramine observed throughout the MD run. Four types of interactions can be distinguished: water bridges (blue), ionic (magenta), hydrophobic (grey), and H-bonds (green). Over the trajectory, the stacked bar charts are normalized. For instance, a value of 0.7 indicates that a particular contact was maintained 70% of the time during simulation. Values greater than 1.0 could occur because a protein residue could interact with the ligand more than once using the same subtype. A timeline explanation of the primary interactions is shown in the following diagram in the figure. Those residues that interacted with the ligand in each trajectory frame are displayed in the output. A darker orange hue denotes several contacts that some residues had with the ligand. The Maestro and Desmond software tools were utilized to generate the pictures (Maestro, Schrödinger LLC, release 2020-3).

**Table 1 biomolecules-14-00610-t001:** MIC and MFC of globospiramine and positive control amphotericin B against *C. albicans* and *C. tropicalis*.

	Globospiramine	Amphotericin B
MIC (µg/mL)		
*C. albicans*	8.0	0.5
*C. tropicalis*	4.0	0.5
MFC (µg/mL)		
*C. albicans*	8.0	0.5
*C. tropicalis*	>64.0	1.0

**Table 2 biomolecules-14-00610-t002:** Binding energies and interactions of globospiramine against *C. albicans* protein targets.

PDB IDs	Globospiramine		Positive Controls					
			Caspofungin		Amphotericin B		Co-Crystallized Ligand/Inhibitor	
	BE (kcal/ mol)	Interactions	BE (kcal/ mol)	Interactions	BE (kcal/ mol)	Interactions	BE (kcal/ mol)	Interactions
1EQP (1,3-β-glucan synthase)	−10.5	His253 (H-bond), Phe258, Phe229 (*pi*–*pi* stacked), Trp277, Val231, Tyr255, Phe229, His253 (*pi*–alkyl), Glu192, His253, Gln230, Glu262 (C-H bond)	−8.0	Trp277, Gln230 (H-bond), Asp227 (salt bridge), His254, Glu262 (C-H bond), Val273, Phe258, Phe144, Tyr255, Trp373 (alkyl, *pi*–alkyl), His253, Arg265 (unfavorable interaction)	−8.8	Asn305, Asp151 (H-bond), Phe258, Phe144 (*pi*–alkyl), Tyr 153 (unfavorable donor–donor, C-H bond)	-	-
4QUV (δ-14-sterol reductase)	−9.5	Arg324, His320 (H-bond), Val96, His320 (*pi*–*sigma*), Leu253, Met99 (alkyl), Arg106, Arg323 (unfavorable positive–positive)	−7.2	Arg106, Arg323, Arg324, Lys406 (H-bond), Tyr414, Trp352, Leu346, Cys403, Trp411, Lys319, Val96 (alkyl, *pi*–alkyl), His320 (*pi*–*pi* stacked), Gln97 (C-H bond), Arg324 (unfavorable positive–positive)	−7.5	Gln97, Glu250, Arg323, Arg324, Gly343 (H-bond), Met99, Leu253 (alkyl), Arg106 (unfavorable positive–positive)	−9.5	His248, Arg313, Thr254, Lys259, Lys319, Trp256, Arg395, Asn316, Thr255 (H-bond), Asp244, Asp399, Arg395 (attractive charge, *pi*–cation), Glu201 (C-H bond), Lys319 (unfavorable positive–positive), Tyr245 (*pi*–*pi* T-shaped), Arg398, Val252 (*pi*–alkyl, alkyl)
5TZ1 (lanosterol 14-alpha demethylase or CYP51)	−7.4	Arg469 (H-bond), Glu444 (attractive charge), Val452, Val454 (alkyl), Ser453, Lys451 (C-H bond)	−5.7	Met508, Pro462, His468, Leu439, Leu471, Gly303 (H-bond), His468 (C-H bond), Ile304 (*pi*–*sigma*), Leu87, Phe233, Tyr64, Phe380, Phe228, Val509, Leu150, Ile304, Ile131, His377, Pro230, Leu88, Lys90 (*pi*–alkyl, alkyl), Arg381, Tyr132, Lys143 (unfavorable interactions)	−3.3	Phe463 (H-bond), His468 (C-H bond), Tyr118 (*pi*–lone pair), Leu376, Ile379, Ala146, Ile304, Leu204, Phe475 (alkyl, *pi*–alkyl), Cys470, Ile379, Gly464, Arg381, Thr311, Phe475, Leu150, Ile471, Tyr132 (unfavorable bonds)	−10.6	Gly303, Ile304 (C-H bond), Ser507, His377 (halogen), Tyr118, Tyr132 (*pi*–*pi*), Leu121, Phe233, Leu376, Pro230, Ile304, Ile131, Lys143 (alkyl, *pi*–alkyl)
5UIV (thymidylate kinase)	−9.4	Gly155, Asp91, Arg39 (H-bond), Asp13, Arg39, Glu159 (*pi*–cation/*pi*–anion/salt bridge), Glu159, Ser18 (C-H bond), Lys17 (*pi*–alkyl)	−8.2	Ser18, Asp13, Asp91, Arg92, Lys17, Lys35, Arg39, Gly157 Gly155 (H-bond), Glu162, Glu159 (salt bridge, attractive charge), Asp13, Asp91, Lys35, Gly155 (C-H bond), Ile196, Arg153, Lys17, Arg39, Val199 (alkyl, *pi*–alkyl)	−7.7	Arg92, Lys35, SerA (H-bond), Glu162, Gln159 (salt bridge, attractive charge), Pro37 (alkyl), Asp13 (C-H bond), Ser18 (unfavorable donor–donor)	−8.9	Arg92, Lys17, Arg14, Ser18, Gly16 (H-bond), Glu159, Asp91, Asp13 (attractive charge, *pi*–anion), Lys35 (unfavorable donor–donor), Tyr100 (*pi*–*pi*), Leu51 (*pi*–alkyl)
4LEB (Als3 adhesin)	−10.6	Thr168 (H-bond), Asp169 (attractive charge), Asp169, Tyr166 (C-H bond), Val161 (*pi*–*sigma*), Val161, Leu167 (alkyl)	−6.5	Thr168, Tyr226, Thr20, Asn22 (H-bond), Trp295 (*pi*–cation), Pro29, Arg171, Tyr21 (alkyl, *pi*–alkyl), Asn22 (*pi*–donor H-bond)	−7.7	Asn22 (H-bond), Tyr226 (*pi*–alkyl), Arg294 (unfavorable positive–positive)	-	-
2Y7L (Als9-2)	−8.1	Thr293 (C-H bond), Trp294 (*pi*–cation), Tyr21, Pro160, Val161 (*pi*–alkyl)	−6.4	Thr168 (H-bond), Arg171, Val22, Pro160, Val161, Ile167, Tyr23, Phe225, Pro29 (alkyl, *pi*–alkyl)	−7.1	Glu86, Ser210, Asn213 (H-bond), Asn211 (C-H bond), Tyr261 (*pi*–alkyl)	-	-

(-) not identified/no co-crystallized ligand attached.

## Data Availability

The original contributions presented in the study are included in the article; further inquiries can be directed to the corresponding author(s).

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
