# Peer review of "Globospiramine Exhibits Inhibitory and Fungicidal Effects against Candida albicans via Apoptotic Mechanisms"

_biomolecules, 2024, doi:10.3390/biom14060610_

Round 1

Reviewer 1 Report

Comments and Suggestions for Authors

The authors of the manuscript titled "Globospiramine exhibits inhibitory and fungicidal effects against Candida albicans via apoptotic mechanisms" explored the efficiency of globospiramine as an antifungal agent. The body of work presented here is appropriate for biomolecules; however, it needs major revisions before it can be considered for publication.

Points that need to be addressed.

  1. To justify the efficacy of Globospiramine as an antifungal agent, the authors need to explore a full panel of fungi strains. The entire data presented in this study only includes only two candida strains which doesn’t guarantee antifungal activity. Please make sure to include Non albicans candida, Candida auris, Aspergillus, Cryptococcus, and other clinical strains.
  2. For the colony forming unit (CFU) viability assay, to explore time- and concentration-dependent activity, the authors explored the activity of the compound for 100 min. Why haven’t the authors explored overnight incubation with the compound to check the cell viability after 24 h.
  3. For the Apoptosis study, there were no significant changes to fluorescence when comparing the vehicle DMSO with the compound at different concentrations. However significant change was observed for AMB. How can you explain apoptosis with this small change and the authors also should explore other markers of apoptosis.
  4. The molecular docking study should be substantiated with experimental results. The authors should explore enzyme inhibition or test cell membrane integrity to validate 1,3-beta-glucan synthase binding.

Author Response

Reviewer 1

The authors of the manuscript titled "Globospiramine exhibits inhibitory and fungicidal effects against Candida albicans via apoptotic mechanisms" explored the efficiency of globospiramine as an antifungal agent. The body of work presented here is appropriate for biomolecules; however, it needs major revisions before it can be considered for publication.

Authors: we thank the referee for the positive evaluation of the work. According to the comments, we submitted a revised version of the manuscript in which the points raised by the reviewer have been addressed. Below we have included a point-by-point response. Furthermore, a tracked version of the manuscript has been submitted to easily check the changes done.

Points that need to be addressed.

To justify the efficacy of Globospiramine as an antifungal agent, the authors need to explore a full panel of fungi strains. The entire data presented in this study only includes only two candida strains which doesn’t guarantee antifungal activity. Please make sure to include Non albicans candida, Candida auris, Aspergillus, Cryptococcus, and other clinical strains.

Authors: Thank you very much for your comment. This study focused on the anticandidal activities of globospiramine against two Candida species (C. albicans and C. tropicalis), which are clinically important species. We have two main reasons as to why these two species were selected. First, the study was conducted to identify the specific bioactive phytoconstituent in anti-C. albicans V. globosa leaf extracts, as reported by Vital & Rivera (2011). In the introduction, we mentioned that this plant has been reported to inhibit C. albicans. Therefore, we used globospiramine for anticandidal investigations against C. albicans. We added C. tropicalis to provide additional insights into its potential effects in non-albicans Candida species, although our study focused on validating the anti-C. albicans activity of the plant (Vital & Rivera 2011). We were able to show in our results that globospiramine exerted in vitro inhibition and putative fungicidal activity vs. C. albicans, validating the earlier claim that V. globosa extracts have anti-C. albicans properties. Second, C. albicans and C. tropicalis are the most commonly used Candida species in anticandidal drug discovery because of their epidemiological relevance. Several recent papers have focused on these two Candida species in drug discovery research (Jafri & Ahmad 2020; Fonseca et al. 2022).

References:

Fonseca, V. J. A., Braga, A. L., de Almeida, R. S., da Silva, T. G., da Silva, J. C. P., de Lima, L. F., ... & Morais-Braga, M. F. B. (2022). Lectins ConA and ConM extracted from Canavalia ensiformis (L.) DC and Canavalia rosea (Sw.) DC inhibit planktonic Candida albicans and Candida tropicalis. Archives of Microbiology, 204(6), 346.

Jafri, H., & Ahmad, I. (2020). Thymus vulgaris essential oil and thymol inhibit biofilms and interact synergistically with antifungal drugs against drug resistant strains of Candida albicans and Candida tropicalis. Journal de mycologie Medicale, 30(1), 100911.

Vital, P. G., & Rivera, W. L. (2011). Antimicrobial activity, cytotoxicity, and phytochemical screening of Voacanga globosa (Blanco) Merr. leaf extract (Apocynaceae). Asian Pacific Journal of Tropical medicine, 4(10), 824-828.

Accordingly, we have changed “antifungal” to “anticandidal” to specify the bioactivity of globospiramine in the entire manuscript wherever necessary. We also added the highlighted portions in the introduction to incorporate the abovementioned rationale in the manuscript:

Revision: As part of our research effort to explore antimicrobial agents from Philippine medicinal plants, as well as to identify potential bioactive phytoconstituents which may contribute to the previously reported anticandidal properties of the plant, we hereby disclose the anticandidal activities of the spirobisindole alkaloid globospiramine (Figure 1) obtained from Voacanga globosa [19] against C. albicans and C. tropicalis using in vitro and computational approaches (molecular docking and molecular dynamics (MD) simulations). 

For the colony forming unit (CFU) viability assay, to explore time- and concentration-dependent activity, the authors explored the activity of the compound for 100 min. Why haven’t the authors explored overnight incubation with the compound to check the cell viability after 24 h.

Authors: Thank you very much for your comment and suggestion. A CFU viability assay was performed to determine the decrease in the CFU count after exposure to the compound. As observed in the graph, there was already a significant decrease at 60 and 90 min of exposure despite this short exposure time. This was observed even in amphotericin B (positive control); therefore, this assay should not be prolonged. Although the decrease in CFU seems to be several folds compared with the 100% baseline in time = 0 min, the CFU viability was not complete at 0% (there were surviving colonies). Therefore, the CFU viability assay was only performed for initial screening of the compound’s anticandidal activity. In the study by Cascio et al. (2013), the CFU count also served as a measure of viability, and decreased CFU viability was observed in just 1-2 h.

The MIC and MFC determination assays, which are considered standard for the assessment of antimicrobial activities of compounds, drugs, and extracts, were performed with an incubation time of 48 h. In the MFC assay, no colony growth was observed in the plates, indicating total cell death. We have revised the manuscript accordingly for clarification.

Reference:

Cascio, V., Gittings, D., Merloni, K., Hurton, M., Laprade, D., & Austriaco, N. (2013). S-Adenosyl-L-Methionine protects the probiotic yeast, Saccharomyces boulardii, from acid-induced cell death. BMC Microbiology13, 1-11.

Revision: 3.1. Effects of globospiramine on C. albicans and C. tropicalis CFU viability

[…..] It is however important to note that although the decrease in % CFU viability seemed to be several folds compared to 100% baseline in time = 0 min, the % CFU viability was not completely at 0% as there were surviving colonies observed. […..]

3.2. MIC and MFC of globospiramine versus C. albicans and C. tropicalis

The significant decrease in CFU viability induced by exposure to globospiramine in a short period of time prompted us to identify its MIC and MFC values, which were identified through a standardized method based on CLSI. […..]

For the Apoptosis study, there were no significant changes to fluorescence when comparing the vehicle DMSO with the compound at different concentrations. However significant change was observed for AMB. How can you explain apoptosis with this small change and the authors also should explore other markers of apoptosis.

Authors: Thank you for your comment. The RFU (relative fluorescence units) were statistically compared with the negative control (vehicle) DMSO. There were significant changes in RFU for 8 and 16 µg/mL of globospiramine (Figure 3A) vs. C. albicans, however, this was not observed in Figure 3B (C. tropicalis). Despite the small change or difference in RFU between the treatment group (8 and 16 µg/mL) and negative control (DMSO), such difference was statistically significant (p<0.05), which is indicative of metacaspase activation in C. albicans. Metacaspase activation is a well-established marker of apoptosis in yeast cells. Studies have reported that targeting metacaspases is expected to have a broad spectrum of action against other fungal pathogens (Kulkarni et al. 2019; Bienvenu et al. 2024). A study also highlighted that activated metacaspases are necessary for optimal growth based on experiments on metacaspase-deficient mutants (Richie et al. 2007). Recent literature has also specified that targeting metacaspases of Candida is considered an emerging therapeutic mechanism of the fungicidal action of anticandidal drugs (Bienvenu et al. 2024). Therefore, we believe that the activation of metacaspases is a logical strategy for assessing the apoptosis-inducing activities of compounds against C. albicans. Overall, our study reports that globospiramine activates C. albicans metacaspase-dependent apoptosis. We have revised some parts of our manuscript to ensure that our work focused on the metacaspase-targeting activity of our compound globospiramine.

References:

Bienvenu, A. L., Ballut, L., & Picot, S. (2024). Specifically Targeting Metacaspases of Candida: A New Therapeutic Opportunity. Journal of Fungi10(2), 90.

Kulkarni, M., Stolp, Z. D., & Hardwick, J. M. (2019). Targeting intrinsic cell death pathways to control fungal pathogens. Biochemical Pharmacology162, 71-78.

Richie, D. L., Miley, M. D., Bhabhra, R., Robson, G. D., Rhodes, J. C., & Askew, D. S. (2007). The Aspergillus fumigatus metacaspases CasA and CasB facilitate growth under conditions of endoplasmic reticulum stress. Molecular Microbiology63(2), 591-604.

Revision: We were very careful in limiting the effect of the compound to metacaspase-inducing activity vs C. albicans. Revisions are highlighted below:

  1. Discussion

[…..] Among the emerging targets in anticandidal drug discovery, targeting metacaspase activation is a promising strategy because of its significant and highly regulated role in apoptosis. Herein, we report the metacaspase-activating potential of globospiramine against C. albicans and C. tropicalis cells using the FAM-FLICA poly-caspase assay. […..] …

[…..] Therefore, further mechanistic investigations on the effect of globospiramine in promoting the growth inhibition of C. tropicalis cells, as well as in other upstream apoptotic pathways for C. albicans are warranted. […..]

The molecular docking study should be substantiated with experimental results. The authors should explore enzyme inhibition or test cell membrane integrity to validate 1,3-beta-glucan synthase binding.

Authors: We acknowledge that such in vitro and/or experimental validation is necessary to validate the proposed targets. Accordingly, the section on computational studies has been revised. Our work determined the mechanism producing the growth inhibition of C. albicans at the cellular level, characterizing the activation of the apoptotic pathway in C. albicans as the main mechanism of globospiramine in significantly reducing the growth of the microorganism. Furthermore, we conducted in silico studies to enlarge the investigation of globospiramine, investigating the potential effects of the compound at the molecular level. We analyzed the possible drug targets of the compound, considering the different targets related to the observed pro-apoptotic effects. Among the selected targets, after in silico analysis, globospiramine could target 1,3-β-glucan synthase and Als3 adhesin. The results were confirmed by MD simulation, which clearly indicated that globospiramine could bind to the target proteins. In addition, considering that targeting the selected enzymes can induce cell apoptosis, it is plausible, given the capacity to induce apoptosis in C. albicans, that the selected enzymes could be inhibited by globospiramine. In fact, after docking studies, only two targets were suitable for globospiramine binding; thus, we hypothesized a possible mechanism of action at the molecular level for globospiramine related to its experimentally determined apoptotic activity on C. albicans. However, to limit the over-interpretation of this part of the study, we revised the discussion, mitigating the wording in this sense, and providing a more careful discussion on the potential targets in the manuscript, including the discussion section. Furthermore, we have added a paragraph in the discussion highlighting the limitations of the in silico approach.

Revision: Although favorable findings were reported in this study, certain limitations need to be considered. In vitro validation of our in silico results is recommended to further confirm the molecular targets of globospiramine in candidal infection pathophysiology. The results of both molecular docking and MD studies provided reliable predictions of how globospiramine could bind to potential target proteins involved in the activation of the apoptotic pathway. It is important that to confirm the pro-apoptotic, fungicidal, and enzyme-inhibitory concentrations, further experiments are requested. Nevertheless, our computational data narrowed down the potential therapeutic targets of globospiramine in the pathophysiology of C. albicans infection.

Reviewer 2 Report

Comments and Suggestions for Authors

The authors present a study examining the inhibitory or fungicidal effects of the plant compound globospiramine against two Candida species. Due to increased numbers of fungal isolates showing resistance against antifungal drugs used in therapy, new compounds are necessary to avoid therapeutic failure. Nevertheless, to establish new drugs is a long way and many compounds were sorted out if they have some unwished properties.

In general:

The compound globospiramine was extracted from the plant Voacanga globosa. Therefore, quality and purity data are necessary for globospiramine. Otherwise, it is difficult to check if the inhibitory and fungicidal effects on Candida albicans or Candida tropicalis depend on the mentioned compound. Are there other minor compounds of the plant extract, which also have an inhibitory effect?

The molecular docking and dynamics simulation represent a putative model and not validated with experimental data like enzyme assays or biofilm formation. The authors may be more careful in their interpretation.

Minor Points

Introduction

Line 46 “Fungal infections…”

Do the authors mean: invasive fungal infections? The most often found pathogen fungi may be skin fungi of the genera Trichophyton or related species.

I miss information how realistic it is, to harvest enough plant material for purification of large amounts of globospiramine in high quality. Is it possible to synthesize the compound?

Materials and Methods

Line 87: “Globospiramine….” See comment under in general.

Line 113: “MIC was recorded” The authors should specify the percentage, does this value mean near MIC50 or near MIC90? The author should specify the value range.

Results

Line 188:”The lowest concentration that significantly…”

As the comment before, the authors should specify more exactly the percentage of MIC values.

Table 1:

In MM the authors mentioned that triplicates were obtained. The authors should include error values for the measured MIC and MFC values

Discussion

This part has to be rewritten.

The molecular docking data are suitable for a model hypothesis of the mode of action for globospiramine but it is no experimental proof.

Without experimental data that support the model hypothesis, the authors must be more carefully with their conclusions and should avoid over interpretation.

Author Response

Reviewer 2

The authors present a study examining the inhibitory or fungicidal effects of the plant compound globospiramine against two Candida species. Due to increased numbers of fungal isolates showing resistance against antifungal drugs used in therapy, new compounds are necessary to avoid therapeutic failure. Nevertheless, to establish new drugs is a long way and many compounds were sorted out if they have some unwished properties.

Authors: we thank the referee for the positive evaluation of the work. According to the comments, we submitted a revised version of the manuscript in which the points raised by the reviewer have been addressed. Below we have included a point-by-point response. Furthermore, a tracked version of the manuscript has been submitted to easily check the changes done.

In general:

The compound globospiramine was extracted from the plant Voacanga globosa. Therefore, quality and purity data are necessary for globospiramine. Otherwise, it is difficult to check if the inhibitory and fungicidal effects on Candida albicans or Candida tropicalis depend on the mentioned compound. Are there other minor compounds of the plant extract, which also have an inhibitory effect?

Authors: We have made the necessary revisions and have included the purity of the compound. In addition, globospiramine is a pure compound. We tested only globospiramine, not the plant extracts or the minor compounds found in the extracts. Thank you.

The molecular docking and dynamics simulation represent a putative model and not validated with experimental data like enzyme assays or biofilm formation. The authors may be more careful in their interpretation.

Authors: we agree with the reviewer, the last response answered to this comments

Minor Points

Introduction

Line 46 “Fungal infections…”

Do the authors mean: invasive fungal infections? The most often found pathogen fungi may be skin fungi of the genera Trichophyton or related species.

Authors: This has been revised to include the word invasive

I miss information how realistic it is, to harvest enough plant material for purification of large amounts of globospiramine in high quality. Is it possible to synthesize the compound?

Authors: Thank you for your clarification. The compound can be synthesized through a complex, multi-step process, as stipulated in the study of Macabeo et al. (2011). In general, synthesis of spirobisindole alkaloids is considered challenging due to the presence of several stereocenters. In addition, globospiramine is the major compound in the V. globosa alkaloidal extract, thus was obtained in large amounts (more than 540 mg) in almost 3.5 kg of plant leaves.

Reference:

Macabeo, A. P. G., Vidar, W. S., Chen, X., Decker, M., Heilmann, J., Wan, B., ... & Cordell, G. A. (2011). Mycobacterium tuberculosis and cholinesterase inhibitors from Voacanga globosa. European Journal of Medicinal Chemistry, 46(7), 3118-3123.

Materials and Methods

Line 87: “Globospiramine….” See comment under in general.

Line 113: “MIC was recorded” The authors should specify the percentage, does this value mean near MIC50 or near MIC90? The author should specify the value range.

Authors: The minimum inhibitory concentration (MIC) was assessed on the basis of the CLSI M27-A3 2008 guidelines on microdilution for yeast species. In this standard protocol, the MIC was recorded as the lowest concentration to induce a significant decrease in yeast cells, not necessarily identifying the percentage of cells inhibited. The decrease can be assessed via spectrophotometry (OD600) or manual visual assessment using the unaided eye based on the turbidity of each well.

Results

Line 188:”The lowest concentration that significantly…”

As the comment before, the authors should specify more exactly the percentage of MIC values.

Authors: In several studies, only the MIC values were reported. MIC value is defined as the minimum concentration that promoted significant decline in number of yeast cells compared to the growth control or the minimum concentration that inhibited the cell growth. This is done either via assessment of changes in OD600 and/or simply by checking turbidity in the wells using the unaided eye (Luephadungphan et al. 2017; Patel et al. 2020; Magpantay et al. 2021).

References:

Kuephadungphan, W., Helaly, S. E., Daengrot, C., Phongpaichit, S., Luangsa-Ard, J. J., Rukachaisirikul, V., & Stadler, M. (2017). Akanthopyrones A–D, α-pyrones bearing a 4-O-methyl-β-D-glucopyranose moiety from the spider-associated ascomycete Akanthomyces novoguineensisMolecules22(7), 1202.

Magpantay, H. D., Malaluan, I. N., Manzano, J. A. H., Quimque, M. T., Pueblos, K. R., Moor, N., ... & Macabeo, A. P. G. (2021). Antibacterial and COX-2 inhibitory tetrahydrobisbenzylisoquinoline alkaloids from the Philippine medicinal plant Phaeanthus ophthalmicus. Plants10(3), 462.

Patel, M., Ashraf, M. S., Siddiqui, A. J., Ashraf, S. A., Sachidanandan, M., Snoussi, M., ... & Hadi, S. (2020). Profiling and role of bioactive molecules from Puntius sophore (Freshwater/brackish fish) skin mucus with its potent antibacterial, antiadhesion, and antibiofilm activities. Biomolecules10(6), 920.

Table 1:

In MM the authors mentioned that triplicates were obtained. The authors should include error values for the measured MIC and MFC values

Authors: MIC values do not necessarily come with error values as they are a product of two-fold dilution and our method of MIC determination was based on significant decrease in cell growth using absorbance analysis (OD600). We selected the lowest concentration that induced significant decrease in cell growth as showed by previous studies (Luephadungphan et al. 2017; Patel et al. 2020; Magpantay et al. 2021). Additionally, the MICs and MFCs were observed in similar concentrations, thus the standard error is equal to zero.

References:

Kuephadungphan, W., Helaly, S. E., Daengrot, C., Phongpaichit, S., Luangsa-Ard, J. J., Rukachaisirikul, V., & Stadler, M. (2017). Akanthopyrones A–D, α-pyrones bearing a 4-O-methyl-β-D-glucopyranose moiety from the spider-associated ascomycete Akanthomyces novoguineensisMolecules22(7), 1202.

Magpantay, H. D., Malaluan, I. N., Manzano, J. A. H., Quimque, M. T., Pueblos, K. R., Moor, N., ... & Macabeo, A. P. G. (2021). Antibacterial and COX-2 inhibitory tetrahydrobisbenzylisoquinoline alkaloids from the Philippine medicinal plant Phaeanthus ophthalmicus. Plants10(3), 462.

Patel, M., Ashraf, M. S., Siddiqui, A. J., Ashraf, S. A., Sachidanandan, M., Snoussi, M., ... & Hadi, S. (2020). Profiling and role of bioactive molecules from Puntius sophore (Freshwater/brackish fish) skin mucus with its potent antibacterial, antiadhesion, and antibiofilm activities. Biomolecules10(6), 920.

Discussion

This part has to be rewritten.

The molecular docking data are suitable for a model hypothesis of the mode of action for globospiramine but it is no experimental proof.

Without experimental data that support the model hypothesis, the authors must be more carefully with their conclusions and should avoid over interpretation.

Authors: We acknowledge that such in vitro and/or experimental validation is necessary to validate the proposed targets. Accordingly, the section on computational studies has been revised. Our work determined the mechanism producing the growth inhibition of C. albicans at the cellular level, characterizing the activation of the apoptotic pathway in C. albicans as the main mechanism of globospiramine in significantly reducing the growth of the microorganism. Furthermore, we conducted in silico studies to enlarge the investigation of globospiramine, investigating the potential effects of the compound at the molecular level. We analyzed the possible drug targets of the compound, considering the different targets related to the observed pro-apoptotic effects. Among the selected targets, after in silico analysis, globospiramine could target 1,3-β-glucan synthase and Als3 adhesin. The results were confirmed by MD simulation, which clearly indicated that globospiramine could bind to the target proteins. In addition, considering that targeting the selected enzymes can induce cell apoptosis, it is plausible, given the capacity to induce apoptosis in C. albicans, that the selected enzymes could be inhibited by globospiramine. In fact, after docking studies, only two targets were suitable for globospiramine binding; thus, we hypothesized a possible mechanism of action at the molecular level for globospiramine related to its experimentally determined apoptotic activity on C. albicans. However, to limit the over-interpretation of this part of the study, we revised the discussion, mitigating the wording in this sense, and providing a more careful discussion on the potential targets in the manuscript, including the discussion section. Furthermore, we have added a paragraph in the discussion highlighting the limitations of the in silico approach.

Revisions: Although favorable findings were reported in this study, certain limitations need to be considered. In vitro validation of our in silico results is recommended to further confirm the molecular targets of globospiramine in candidal infection pathophysiology. The results of both molecular docking and MD studies provided reliable predictions of how globospiramine could bind to potential target proteins involved in the activation of the apoptotic pathway. It is important that to confirm the pro-apoptotic, fungicidal, and enzyme-inhibitory concentrations, further experiments are requested. Nevertheless, our computational data narrowed down the potential therapeutic targets of globospiramine in the pathophysiology of C. albicans infection.

Round 2

Reviewer 1 Report

Comments and Suggestions for Authors

Satisfactory improvements are made to improve the quality of the manuscript

Reviewer 2 Report

Comments and Suggestions for Authors

Congrats, my comments are answered succesfully